# Emotional Dysregulation, Anxiety Symptoms and Insomnia in Individuals with Alcohol Use Disorder

**DOI:** 10.3390/ijerph19052700

**Published:** 2022-02-25

**Authors:** Dorota Wołyńczyk-Gmaj, Andrzej Jakubczyk, Elisa M. Trucco, Paweł Kobyliński, Justyna Zaorska, Bartłomiej Gmaj, Maciej Kopera

**Affiliations:** 1Department of Psychiatry, Medical University of Warsaw, 02-091 Warsaw, Poland; dorota@psych.waw.pl (D.W.-G.); ajakubczyk@wum.edu.pl (A.J.); just.zaorska@gmail.com (J.Z.); maciej.kopera@wum.edu.pl (M.K.); 2Department of Psychology, Center for Children and Families, Florida International University, Miami, FL 33199, USA; etrucco@fiu.edu; 3Department of Psychiatry, Addiction Center, University of Michigan, Ann Arbor, MI 48109, USA; 4Laboratory of Interactive Technologies, National Information Processing Institute, 00-608 Warsaw, Poland; pawel.kobylinski@opi.org.pl

**Keywords:** anxiety, insomnia, alcohol use disorder, emotional dysregulation, emotions, alcohol dependence

## Abstract

Alcohol craving is associated with insomnia symptoms, and insomnia is often reported as a reason for alcohol relapse. The current study examined associations between emotional regulation, anxiety, and insomnia among a group of 338 patients with alcohol use disorder (AUD). Because insomnia most often develops after stressful experiences, it was expected that anxiety symptoms would mediate the association between emotional dysregulation and insomnia severity. It was also expected that an insomnia diagnosis would moderate the association between emotional dysregulation and anxiety symptoms, namely that higher anxiety levels would be found in individuals with insomnia than in those without insomnia. Insomnia severity was assessed with a total score based on the Athens Insomnia Scale (AIS). Additionally, an eight-point cut-off score on the AIS was used to classify participants as with (*n* = 107) or without (*n* = 231) an insomnia diagnosis. Moreover, participants completed the Emotion Regulation Scale (DERS; total score) and the Brief Symptoms Inventory (BSI; anxiety). Individuals with insomnia did not differ from those without insomnia in age (*p* = 0.86), duration of problematic alcohol use (*p* < 0.34), mean days of abstinence (*p* = 0.17), nor years of education (*p* = 0.41). Yet, individuals with insomnia endorsed higher anxiety (*p* < 0.001) and higher emotional dysregulation (*p* < 0.001). Anxiety symptoms fully mediated the association between emotional dysregulation and insomnia severity (*p* < 0.001). Furthermore, insomnia diagnosis positively moderated the association between emotional dysregulation and anxiety (*p* < 0.001). Our results suggest that emotional dysregulation can lead to insomnia via anxiety symptoms. Treating anxiety symptoms and emotional dysregulation could help to prevent or alleviate symptoms of insomnia in people with AUD. Moreover, treating insomnia in people with AUD may also have a positive effect on anxiety symptoms.

## 1. Introduction

Insomnia often co-occurs with alcohol use disorder (AUD). The prevalence of insomnia symptoms in individuals with AUD ranges from 31% up to 91%, depending on the definition of insomnia and sample characteristics [1,2,3,4,5,6,7,8,9]. In Poland, 62.9% of individuals with AUD report symptoms of insomnia, especially long sleep latency [10]. In Poland, the prevalence of insomnia is about 28% in the general population [11,12]. Thus, insomnia within the AUD population is significantly higher in comparison to the general Polish population.

A key feature of insomnia is twenty-four-hour hyperarousal, which is characterized by chronic activation of both limbs of the stress system (i.e., the hypothalamic–pituitary–adrenal axis (HPA) and the sympathetic system), hypersecretion of adrenocorticotropic hormone (ACTH) and cortisol [13,14], elevated metabolic rates [15], and greater global cerebral glucose metabolism during nighttime and wakefulness [16,17,18]. Importantly, hyperarousal also commonly accompanies AUD. Although acute alcohol withdrawal commonly lasts about one week, insomnia among individuals with AUD could last for several months, which is well-recognized as a risk factor for relapse [19]. Self-reported subjective sleep problems, as well as more objective changes identified using polysomnography (PSG), especially in rapid eye movement (REM) sleep, were also supported as significant prospective markers of relapse [20,21]. PSG studies have indicated that characteristic changes in electroencephalography, such as longer REM sleep times, shorter REM sleep latency, longer non-REM (NREM) 1 stage time, shorter NREM 3 stage time, and decreased slow-wave activity during NREM, are present after 30–179 days of abstinence [22] and could persist up to 2 years after an individual’s last drink [23]. Importantly, insomnia commonly precedes the development of AUD [24]. Accordingly, insomnia (similar to other psychiatric disorders) may represent a risk factor for AUD [25].

Anxiety is considered one of the major precipitating factors of insomnia. Moreover, insomnia is commonly present in anxiety disorders, especially general anxiety disorder (GAD) and post-traumatic stress disorder (PTSD). Notably, GAD is most commonly comorbid with insomnia [26], with difficulty falling asleep and frequently waking up at night being the most characteristic manifestations of anxiety [27,28]. According to Harvey’s cognitive model for insomnia, the anxious state is responsible for insomnia [29]. On the other hand, the Norwegian population-based cohort study that focused on data from two general health surveys performed 11 years apart showed significant relations between the longitudinal course of chronic insomnia and the development of anxiety disorders. Thus, people with chronic insomnia are more likely to develop anxiety over time [30].

Anxiety is a common symptom described in individuals with AUD. Grant et al. observed a positive association between anxiety symptoms and severity of alcohol dependence [31]. Moreover, anxiety symptoms are considered a significant risk factor for relapse [31,32]. In the general population, those who report a short sleep duration (6 h or less) are more likely to be diagnosed with anxiety disorders (i.e., panic disorder, PTSD, and generalized anxiety disorder), in addition to AUD, than those who report a longer sleep duration [33].

Anxiety is also associated with difficulties in emotional regulation [34]. Tsypes and colleagues demonstrated that emotional dysregulation mediated the association between anxiety and insomnia among patients diagnosed with GAD [35]. It was suggested that anxiety severity may be used to determine whether individuals with emotional dysregulation go on to develop insomnia. Clinical symptoms of insomnia usually appear after stressful experiences or as a result of common life problems [36]. In turn, these experiences trigger excessive stress axis activity, manifested by hyperarousal [18,37], anxiety symptoms, and difficulty falling asleep and/or maintaining sleep, which is perpetuated by the patients in the course of inappropriate strategies for coping with insomnia [38]. It is thought that individuals with poor emotional regulation are more likely to experience poor mood, intrusive thoughts, ruminations, and anger when experiencing stress [39]. Thus, their sleep problems increase, which can develop into insomnia. Importantly, not all psychiatric disorders characterized by poor emotional regulation are also characterized by insomnia. Yet, individuals with AUD are characterized by both emotional dysregulation and high levels of anxiety, but their mutual effect on insomnia has not been thoroughly studied in this sample [40,41]. 

Current research supports the strong contribution of emotional dysregulation in the etiology of substance use [42] to the extent that substance use disorders—including AUD—have repeatedly been conceptualized as disorders of emotional regulation [43]. Dysregulation of emotions is found among those with insomnia as well. Low resilience to stressful events and a poor adaptational capacity are considered predisposing factors of insomnia. On the other hand, sleep disorders can negatively affect emotional regulation because sleep is involved in prefrontal control of emotions. Sleep deprivation is associated with increased limbic system activity and a corresponding lack of prefrontal regulation over limbic activation [44]. During REM sleep, limbic system structures are activated [45] and disruption of this stage of sleep among individuals with insomnia can amplify a state of hyperarousal. From a clinical perspective, patients with insomnia often report irritability and explosiveness—typical symptoms of emotional dysregulation [46].

Taken together, previous work suggests that dysregulation of emotions and anxiety symptoms may increase the risk of insomnia, especially among patients with AUD. In addition, these factors (i.e., emotional dysregulation, anxiety, and insomnia), are consistently considered to increase relapse among individuals with AUD. Additionally, insomnia disturbs regulation of emotion and contributes to the development of anxiety disorders. The aim of the current study was to examine whether emotional dysregulation and anxiety symptoms would increase the severity of insomnia among individuals with AUD. Given possible bidirectional associations between these variables, we ultimately chose insomnia as an outcome in our analysis, as sleep problems are the most common reason for patient referral to a specialist [47]. We predicted that anxiety symptoms would mediate the association between emotional dysregulation and insomnia severity. Yet, it was also hypothesized that the association between anxiety symptoms and emotional dysregulation would differ across those with and without an insomnia diagnosis. We expected that insomnia diagnosis would moderate the interaction between emotional dysregulation and anxiety symptoms, specifically, that the effect of emotional dysregulation on anxiety symptoms would be stronger among individuals with co-occurring insomnia and AUD given greater hyperarousal. 

## 2. Materials and Methods

### 2.1. Participants

The sample was taken from an ongoing study examining emotional and behavioral functioning among individuals with AUD who are being treated in an inpatient setting. The sample for this study included 338 adults (42.8 ± 10.7 years of age) admitted to an eight-week, drug-free (without any benzodiazepine use), abstinence-based, inpatient alcohol treatment program. The treatment program was designed for individuals who had a period of abstinence, did not exhibit acute withdrawal symptoms, and were referred from either home or detoxification facilities. Research procedures were conducted during the first two weeks after patients were admitted for treatment. This ensured at least two weeks of complete abstinence from alcohol; yet, the reported period of severity was significantly longer (see Table 1). Given the overrepresentation of men in substance use treatment programs in Poland, a majority of the sample consisted of white men (83%). 

AUD diagnosis was assessed using the International Classification of Diseases and Related Health Problems 10th Revision upon treatment admission and was then verified with the MINI International Neuropsychiatric Interview. Exclusion criteria included the following: a clinically significant cognitive deficit (<25 on the Mini-Mental State Examination), a history of psychosis, co-occurring current psychiatric disorders requiring medication, current use of analgesics, and co-occurrence of substance use/dependence other than nicotine. 

All study procedures were conducted in accordance with the ethical principles described in the Declaration of Helsinki in 1964 and received approval from the Bioethics Committee of the institution where the study took place. The study protocol did not include a follow-up assessment of relapse, sleep problems, or any other psychopathological variables.

### 2.2. Measures

Sociodemographic information. Questions regarding sociodemographic characteristics of the sample (e.g., age, biological sex, education) were obtained using a self-report questionnaire.

Alcohol use factors. The Short Inventory of Problems [48] was used to quantify the number of consecutive days of heavy drinking, maximum amount of alcohol consumed during periods of consecutive heavy drinking, and the period of abstinence prior to the assessment via an interview. The modified version of the Substance Abuse Outcomes Module [49] included an item regarding the onset age of drinking problems, which was used to determine the duration of problematic alcohol use.

Emotional dysregulation. Emotional dysregulation was assessed using the Polish version of the Difficulties in Emotional Regulation Scale (DERS) [50,51]. The DERS comprises 36 items scored on a 5-point scale (1 = almost never to 5 = almost always). The total DERS score ranges from 36 to 180 with higher scores reflecting greater difficulties regulating emotion. A total DERS score was used for the current study (Cronbach’s alpha = 0.93). 

Anxiety. The anxiety subscale from the Brief Symptom Inventory—BSI [52] was used to measure current anxiety symptoms. 

Insomnia severity and diagnosis. The Athens Insomnia Scale (AIS) was used to assess both insomnia severity and diagnosis. The AIS is an 8-item questionnaire based on the International Classification of Diseases, 10th Revision criteria designed for quantitative measurement of insomnia. Each item is scored on a 4-point scale (0 = not a problem to 3 = a very serious problem). The total score can range from 0 to 24. The scale has very good consistency (Cronbach alpha = 0.88) and reliability (test–retest reliability, *r*^2^ = 0.92). The AIS is one of the most commonly used scales for diagnostic purposes as well as research on the effectiveness of insomnia treatment. The total AIS score was used to assess insomnia severity on a continuous scale. Additionally, the AIS was also used to determine insomnia diagnosis consistent with prior work [53]. Specifically, an 8-point diagnostic cut-off score was used to classify individuals as either “with” (*n* = 107) or “without” (*n* = 231) a diagnosis of insomnia.

### 2.3. Data Analysis

In order to determine the most important variables differentiating those with and without insomnia in individuals with AUD, emotional dysregulation, anxiety severity, age, and biological sex were included in one-way ANOVA between-group comparisons. 

#### 2.3.1. Mediation Model

In order to test anxiety symptoms (BSIANX score) as a potential mediator in the association between emotional dysregulation (DERS TOT score) and insomnia (AIS total score), [53] the PROCESS SPSS macro for mediation analysis with bootstrapping (5000 resamples with replacement) was applied (see Figure 1 for the conceptual model). Biological sex and age were included as covariates. 

#### 2.3.2. Moderation Model

Subsequently [53], the PROCESS SPSS macro for moderation analysis with bootstrapping (5000 resamples with replacement) was applied in order to test insomnia diagnosis (based on the AIS cutoff score) as a potential moderator in the association between emotional dysregulation (DERS TOT score) on anxiety symptoms (BSIANX score). Biological sex and age were included as covariates (see Figure 2 for the conceptual model). Next, a simple slope analysis reflecting the “pick-a-point” approach was conducted to probe the significant interactions. This consisted of a set of regression tests (two in this case) that determined where in the distribution of the moderator the predictor influenced the dependent variable [54]. Nonstandardized coefficients are reported throughout the paper.

## 3. Results

A description of the between-group differences in individuals with AUD (with and without insomnia) based on sociodemographic information and clinical and alcohol use characteristics is provided in Table 1.

### 3.1. Mediation Model

The mediation model with emotional dysregulation (DERS TOT score) as the predictor, anxiety symptoms (BSIANX score) as the mediator, and insomnia (AIS total score) as the dependent variable was tested with biological sex and age as covariates (see Figure 3 for nonstandardized coefficients). The results supported mediation. The model explained 32% of the variance in anxiety (R^2^ = 0.318; F(3, 250) = 38.818; *p* < 0.001) and 33% of the variance in in insomnia (R^2^ = 0.330; F(4, 249) = 30.676; *p* < 0.001). More specifically, greater emotional dysregulation led to greater anxiety (b = 0.019; 95% CI = [0.015, 0.023]; *p* < 0.001), which in turn resulted in greater insomnia severity (b = 3.223; 95% CI = [2.440, 4.006]; *p* < 0.001). Although the direct effect of emotional dysregulation on insomnia was significant (i.e., greater emotional dysregulation resulted in more severe insomnia), the indirect effect via anxiety symptoms was also significant (c = 0.061; bootstrap 95% CI = [0.042, 0.081]). This indicates that anxiety may represent one potential pathway linking emotional dysregulation to insomnia severity. 

### 3.2. Moderation Model

The moderation model with emotional dysregulation as the predictor, insomnia diagnosis (AIS cutoff score) as the moderator, and anxiety symptoms (BSIANX score) as the dependent variable was tested with biological sex and age as covariates (see Figure 4 for nonstandardized coefficients). The model explained 44% of the variance in anxiety symptoms (*R*^2^ = 0.441; *F*(5, 248) = 39.191; *p* < 0.001). A significant interaction was found between emotional dysregulation and insomnia diagnosis (∆*R*^2^
*=* 0.024; *F*(1, 248) = 10.585; *p* = 0.001). Findings from probing the interaction (Figure 5 indicate that the simple slopes for the regression of anxiety symptoms on emotional dysregulation were statistically significant for individuals both with insomnia and without an insomnia diagnosis. Both slopes were positive, yet the slope for those with insomnia (*b* = 0.023; 95% CI = [0.017, 0.029] *p* < 0.001) was twice as steep as the slope for those without insomnia (*b* = 0.011; 95% CI = [0.007, 0.015]; *p* < 0.001). That is, one standard deviation increase in emotional dysregulation was associated with almost two times greater anxiety symptom severity among individuals with insomnia compared to those without insomnia. 

## 4. Discussion

To the best of our knowledge, this is the first study testing the mediating effect of anxiety in the association between emotional dysregulation and insomnia severity. We also examined the synergistic effect of emotional dysregulation and insomnia diagnostic status on anxiety in a sample of individuals with AUD. Findings indicate that anxiety mediates the association between emotional dysregulation and insomnia severity. Specifically, higher emotional dysregulation was associated with more anxiety, which in turn predicted greater insomnia severity. These results suggest that anxiety may represent a potential mechanism linking emotional dysregulation and insomnia among individuals with AUD. Moreover, insomnia diagnosis status was found to significantly moderate the association between emotional dysregulation and anxiety. Specifically, the same level of emotional dysregulation was related to almost two times greater anxiety symptom severity among those with insomnia compared to those without insomnia. 

The Insomnia 3 P model [55] has utility in framing our findings within the broader context. The Insomnia 3 P model, also known as the Spielman model, describes how acute insomnia develops and can progress to a chronic and self-perpetuating syndrome. It outlines three key factors that are linked to insomnia: predisposing, precipitating, and perpetuating factors. Predisposing factors reflect personality traits, temperament, poor socioeconomic status, and genetic vulnerability. Stressful life events and somatic disease are the most common precipitating factors. Ineffective coping strategies, stressful life events, and poor health status can perpetuate insomnia. With respect to the current study, emotional dysregulation represents a predisposing trait factor that may lead to low resilience to stress and increased anxiety. In turn, anxiety, as manifested by cognitive pre-sleep hyperarousal, likely perpetuates insomnia [56]. Moreover, alcohol withdrawal syndrome is believed to stem from high levels of anxiety and hyperarousal [57]. Thus, these results suggest that among individuals with AUD, co-occurring anxiety symptoms and emotional dysregulation likely increase the risk for the development of insomnia. Yet, reciprocal effects are also in play, whereby insomnia drives dysregulation of emotion and anxiety symptoms, which characterizes the self-perpetuating nature of insomnia. 

Support for the association between insomnia, anxiety, and emotional dysregulation among individuals with AUD (N = 220) and/or use of other psychoactive substances has already been demonstrated in prior work [58]. That is, insomnia and emotional dysregulation were strongly correlated with anxiety, depression, PTSD, and alcohol drinking severity [58]. Subsequently [41] investigated insomnia correlates in a sample of college students. They found that among a subsample of participants who met the criteria for insomnia (*n* = 136), there was evidence that insomnia was positively correlated with anxiety symptoms, but only for those high in emotional dysregulation. Among participants low in emotional dysregulation, anxiety symptom severity did not impact subjective sleep quality [41]. Finally, Tsypes and colleagues showed that among individuals with GAD, emotional dysregulation mediated the association between anxiety and insomnia severity [35]. Thus, greater difficulties with emotional regulation increased the risk of insomnia symptoms among those high in anxiety symptomatology. Our study represents a slightly different approach, with anxiety (not emotional regulation) being a mediating factor. From a clinical perspective, we perceive emotional regulation as a stable trait-related factor, while anxiety reflects a state-related factor, which is prone to greater fluctuations based on life stressors or withdrawal symptoms among individuals with AUD. Accordingly, it follows that emotional regulation represents a precipitant of anxiety and is modeled as such in the estimated model.

Prior work supports a bidirectional association between sleep and regulation of emotions; thus, insomnia could also contribute to emotional dysregulation [59]. During sleep, emotional and reward networks are activated to optimize emotional responses during the day [60]. Sleep disruption can negatively influence decision-making by modulating brain structure activity, especially within the nucleus accumbens and insula. Moreover, among individuals with insomnia, prefrontal cortex activity pertaining to supervisory control, problem solving, flexibility, and self-control via the regulation of amygdala reactivity is often impaired [61]. Thus, insomnia relates to impaired regulation of emotions and may increase the tendency to engage in risky, impulsive, and aggressive behaviors [62] and could predict emotional dysregulation. A specific aspect of sleep problems, poor sleep quality, was found to negatively impact emotional regulation [63]. Still, impaired emotional regulation was found to increase hyperarousal, which led to reports of sleep disruption in the subsequent night [63]. Therefore, in our study, we also examined the effect of insomnia diagnosis status on the relationship between emotional dysregulation and anxiety.

Our findings indicate that emotional dysregulation increases anxiety, and anxiety induces insomnia in people with AUD. Yet, there are other possible reasons that may explain this association, such as a direct toxic effect of alcohol withdrawal, a tendency for insomnia to induce both anxiety and emotional dysregulation, and the possibility of hidden confounding variables (e.g., genes) underlying all of these factors.

Consistently, individuals with AUD are characterized by high emotional dysregulation, increased levels of anxiety, and insomnia. Moreover, poor emotional regulation in stressful situations or during periods of alcohol withdrawal often results in severe anxiety symptoms and further precipitation of insomnia. Insomnia, in turn, worsens the regulation of emotions and increases anxiety and and alcohol craving. Therefore, insomnia increases the severity of AUD. All of these factors (i.e., insomnia, anxiety, and emotional dysregulation) should be considered as treatment targets among patients with AUD. Addressing sleep problems could help to prevent the vicious cycle worsening the mental state of patients with AUD by reducing anxiety and enhancing more effective emotional regulation. Interventions aimed at minimizing sleep problems in the early stages of AUD treatment could prevent emotional dysregulation, alcohol craving, and relapse, especially in patients demonstrating high levels of anxiety. For example, effective treatments for insomnia, such as trazodone [64] and/or cognitive-behavior therapy for insomnia (CBT-I) [65], should be considered when treating patients with co-occurring AUD and insomnia. On the other hand, reducing anxiety may also have utility in managing insomnia.

Our results have several limitations. First, we did not assess sleep with objective sleep measures (e.g., PSG). However, insomnia as a disorder is often diagnosed based on the subjective experience of patients and not PSG results. In addition, study results may not generalize to other samples of AUD patients. Our study group may differ in terms of psychopathological characteristics (i.e., more severe sleep problems, emotional dysregulation, and anxiety) compared to individuals with AUD who fail to seek treatment. Moreover, men were overrepresented in the study sample (83%), reducing the generalizability of the results for both sexes. Additionally, because the relationship between insomnia, anxiety, and emotional dysregulation is bidirectional, an adequate model accounting for these reciprocal associations could not be constructed. Future work that can tease apart how these three constructs unfold over time is necessary.

## 5. Conclusions

In conclusion, our study indicates that high emotional dysregulation and high anxiety severity may increase the risk of developing insomnia among patients with AUD. Specifically, anxiety symptoms mediate the association between emotional dysregulation and insomnia. Carrying an insomnia disorder diagnosis may also increase the level of anxiety experienced among individuals with AUD. Routinely evaluating sleep problems both subjectively and objectively may offer clinical utility when treating individuals with AUD. Moreover, it is likely that addressing co-occurring emotional dysregulation and/or anxiety may prevent insomnia among individuals with AUD to enhance treatment outcomes. 

## Figures and Tables

**Figure 1 ijerph-19-02700-f001:**
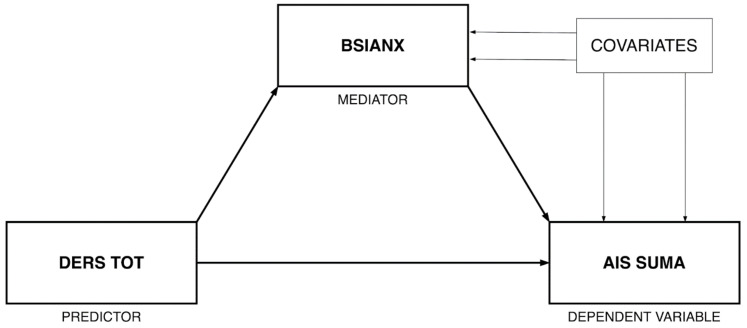
Mediation model conceptual diagram.

**Figure 2 ijerph-19-02700-f002:**
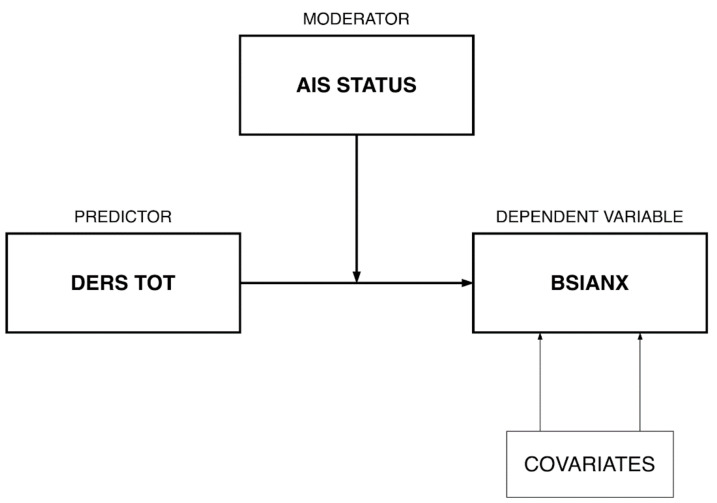
Moderation model conceptual diagram.

**Figure 3 ijerph-19-02700-f003:**
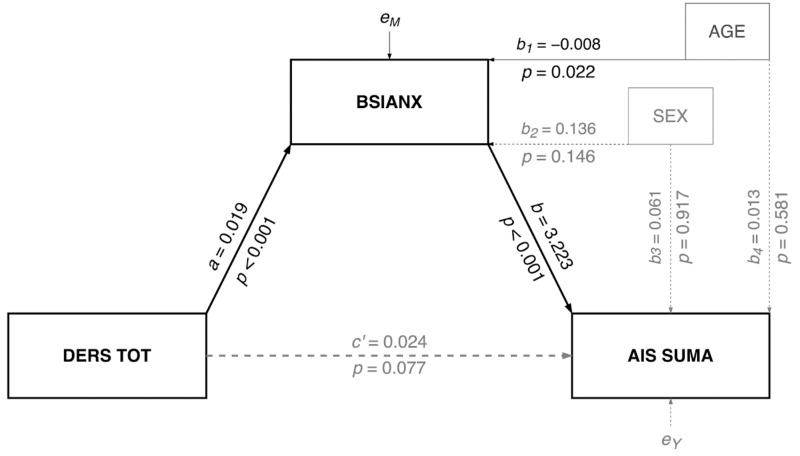
Mediation model statistical diagram.

**Figure 4 ijerph-19-02700-f004:**
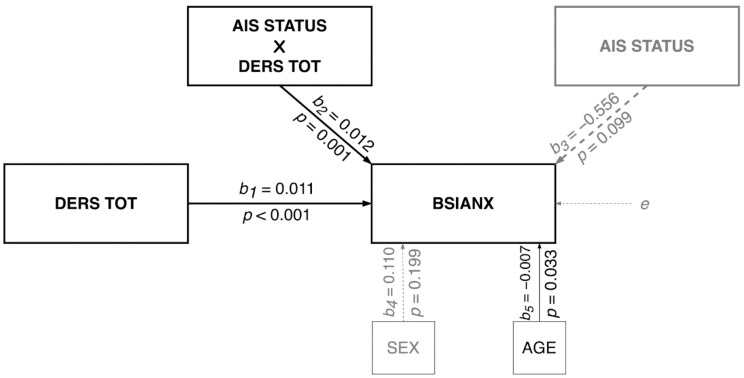
Moderation model statistical diagram.

**Figure 5 ijerph-19-02700-f005:**
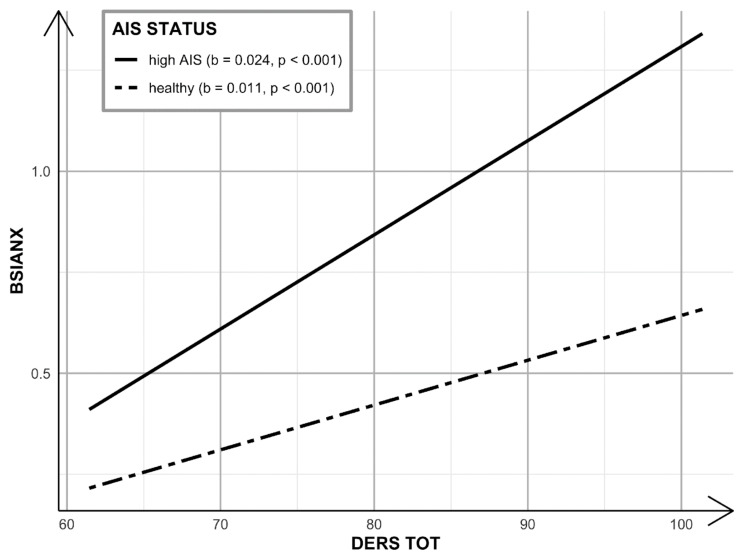
Moderation model interaction probing.

**Table 1 ijerph-19-02700-t001:** Socio-demographic and clinical characteristics of the individuals with AUD with and without insomnia.

	Individuals with Insomnia	Individuals without Insomnia	*p* ^a^
Age (years) ^b^	43.04 (10.59) (N = 107)	42.73 (10.59) (N = 231)	0.81
Education (years) ^b^	12.97 (3.31) (N = 106)	13.35 (4.17) (N = 231)	0.41
Biological sex (man/woman)	88/19 (N = 107)	195/36 (N = 231)	0.36
Severity of anxiety symptoms ^c^	1.12 (0.89) (N = 107)	0.48 (0.56) (N = 231)	<0.001
Duration of problematic alcohol use (years) ^b^	24.83 (34.55) (N = 75)	12.15 (14.01) (N = 179)	0.34
Mean abstinence length (days) ^a^	71.21 (91.43) (N = 66)	104.13 (178.13) (N = 94)	0.17
Emotional dysregulation	91.88 (19.52) (N = 75)	77.02 (18.49) (N = 179)	<0.001
Severity of insomnia	11.74 (3.21) (N = 107)	3.71 (2.08) (N = 231)	<0.001

^a^ Analysis of variance or Mann–Whitney tests were applied to test differences between alcohol-dependent patients and healthy controls for variables that were either normally distributed or not, respectively. ^b^ Normally distributed variable described by mean (*SD*). ^c^ Non-normally distributed variable described by median (interquartile range). ^d^ Percent of individuals scoring high (>61) on the Toronto Alexithymia Scale.

## Data Availability

Data are available on request from the corresponding author.

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
