# Peer review of "Emotional Dysregulation, Anxiety Symptoms and Insomnia in Individuals with Alcohol Use Disorder"

_ijerph, 2022, doi:10.3390/ijerph19052700_

Round 1

Reviewer 1 Report

The justification for the importance of insomnia in the population with alcohol use disorder is well done. The authors correctly describe the physiological changes that occur during the sleep of individuals with AUD and how insomnia is a risk factor for a relapse.

Likewise, the close relationship between insomnia and anxiety is well related theoretically, and especially in AUD population. The authors correctly justify, based on the literature, the relationship between emotional dysregulation, anxiety and insomnia, as well as the adverse effects of insomnia on anxiety and emotional dysregulation.

From this theoretical justification, they formulate their hypotheses correctly.

Regarding the study sample, in my opinion, it is adequate in terms of size, and appropriate inclusion/exclusion criteria have been used. It's a pity it's not matched in terms of sex (83% men).

The instruments used are appropriate for the purpose of the work. The data analyses carried out are adequate to test the hypotheses of the work. Regarding the results they are very well exposed, in a short, clear and concise way, as it should be.

Regarding the discussion, the authors explain their contributions according to previous studies, taking into consideration not only the results of measures of psychological variables but also the physiological reasons that provoke it. The authors explain their results clearly according to the literature. This is very suitable. The authors are adequately cautious in their claims.

I do not consider the absence of an objective measure of sleep to be a limitation of work. It is more interesting and appropriate, what the authors have done, to measure the subjective quality of sleep.

The conclusions are concise and prudent. They are well explained.

Author Response

We now state the overrepresentation of men as a study limitation (see page 10)

Reviewer 2 Report

In my opinion this article is sufficiently novel and interesting. Presented article brings new, meaningful information. Alcoholism has an enormous impact on public health, and  Alcohol use disorder (AUD) is a very common and very complex disease. It is highly important to find out the new ways of treatment and to broaden the knowledge about it. Study presented in this article is very valuable. Deep value of presented study is the analyses of complicated groups of mental disorders such as insomnia, anxiety and emotional dysregulations in these very particular group od patients.

 I strongly suggest acceptation for publication in International Journal of Environmental Research and Public Health, after very small correction pointed out in Comments for the Authors.

In the Bibiography part there some incorrect number showed up before number of 38 ( line 434).

Author Response

We have made the necessary corrections to the References section

Reviewer 3 Report

The authors present an interesting descriptive cross-sectional study assessing anxiety and insominia in patients with AUD.

Because a key symptom of alcohol withdrawal in patients with AUD is anxiety, the assessement of anxiety in patients with AUD is difficult. But the strength of this particular study is that it was conducted in an inpatient setting, in patients with severe AUD, yet all in a state of documented abstinence.

For the reader, it would be interesting to read some more information in the method section:

  • when was the assessment performed? was the delay of two weeks between entrey and assessement equivalent to the delay between alcohol cessation and the interview standardized?
  • were the patients referred from a an accute in patient setting to maintain abstinence in this facility or could they arrive directly from home and experiment a withdrawal syndrome in the first few days of the the stay?
  • are some patients receiving medication such as benzodiazepine in the first few days of the present stay?
  • could some of them still experiment withdrawal synmptoms at the time of assessment?
  • do the authors have collected longitudinal data on the duration of the insomnia ?
  • do the authors have collected datat on  the outcome of the stay or relapse?

in the introduction the authors could cite this article: PMID 321140 30 that shows that both anxiety disorders and AUD are associated with shorter duration of sleep in the general population.

in the discussion, the authors could cite the 4 possible reasons for this association of emotion dysregulation and insomnia in AUD: a direct toxic effect of alcohol withdrawal, the pathway that they describe as fitting well with their data (emotion dysregulation induces anxiety induces insomnia), a tendency to insomnia inducing both anxiety and  emotion dysregulation and insomnia,  or the possibility of hidden confounding factors associated with all those conditions (such as genetic factors).

overall an interesting cross-sectional study.

Author Response

For the reader, it would be interesting to read some more information in the method section:

when was the assessment performed? was the delay of two weeks between entry and assessment equivalent to the delay between alcohol cessation and the interview standardized?

were the patients referred from a an acute in patient setting to maintain abstinence in this facility or could they arrive directly from home and experiment a withdrawal syndrome in the first few days of the stay?

are some patients receiving medication such as benzodiazepine in the first few days of the present stay?

could some of them still experiment withdrawal symptoms at the time of assessment?

do the authors have collected longitudinal data on the duration of the insomnia?

do the authors have collected data on the outcome of the stay or relapse?

We have revised the Methods section to incorporate this additional information as requested by the reviewer. The following is now stated in the manuscript: “The sample for this study included 338 adults (42.8 ± 10.7 years of age) admitted to an eight-week, drug-free (without any benzodiazepine use), abstinence-based, inpatient alcohol treatment program. The treatment program was designed for individuals who had a period of abstinence, did not exhibit acute withdrawal symptoms, and were referred from either home or detoxification facilities. Research procedures were conducted during the first two weeks after patients were admitted for treatment. This ensured at least two weeks of complete abstinence from alcohol; yet, reported period of severity was significantly longer (see Table 1). Given the overrepresentation of men in substance use treatment programs in Poland, a majority of the sample consisted of White men (83%) (…). The study protocol did not include a follow-up assessment of relapse, sleep problems, or any other psychopathological variables” (pages 3-4).

In the introduction the authors could cite this article: PMID 321140 30 that shows that both anxiety disorders and AUD are associated with shorter duration of sleep in the general population.'

Thank you for the valuable suggestion. We now discuss this work in the Introduction: “…In the general population, those who report a short sleep duration (6 hours or less) are more likely to be diagnosed with anxiety disorders (i.e., panic disorder, PTSD, and generalized anxiety disorder), in addition to AUD, than those who report a longer sleep duration [33]...” (pages 2-3).

'in the discussion, the authors could cite the 4 possible reasons for this association of emotion dysregulation and insomnia in AUD: a direct toxic effect of alcohol withdrawal, the pathway that they describe as fitting well with their data (emotion dysregulation induces anxiety induces insomnia), a tendency to insomnia inducing both anxiety and  emotion dysregulation and insomnia,  or the possibility of hidden confounding factors associated with all those conditions (such as genetic factors).

We thank the reviewer for this thoughtful suggestion. We have edited the manuscript to incorporate these four possible explanations: "Our findings indicate that emotional dysregulation increases anxiety, and anxiety induces insomnia in people with AUD. Yet, there are other possible reasons that may explain this association, such as: a direct toxic effect of alcohol withdrawal, a tendency for insomnia to induce both anxiety and emotion dysregulation, and the possibility of hidden confounding variables (e.g., genes) underlying all of these factors.” (page 10).